# Distinct submembrane localisation compartmentalises cardiac NPR1 and NPR2 signalling to cGMP

Hariharan Subramanian[1,2], Alexander Froese[1,2,3], Peter Jönsson [4], Hannes Schmidt[5], Julia Gorelik[6] & Viacheslav O. Nikolaev [1,2]

Natriuretic peptides (NPs) are important hormones that regulate multiple cellular functions including cardiovascular physiology. In the heart, two natriuretic peptide receptors NPR1 and NPR2 act as membrane guanylyl cyclases to produce 3′,5′-cyclic guanosine monophosphate (cGMP). Although both receptors protect from cardiac hypertrophy, their effects on contractility are markedly different, from little effect (NPR1) to pronounced negative inotropic and positive lusitropic responses (NPR2) with unclear underlying mechanisms. Here we use a scanning ion conductance microscopy (SICM) approach combined with Förster resonance energy transfer (FRET)-based cGMP biosensors to show that whereas NPR2 is uniformly localised on the cardiomyocyte membrane, functional NPR1 receptors are found exclusively in membrane invaginations called transverse (T)-tubules. This leads to far-reaching CNP/NPR2/cGMP signals, whereas ANP/NPR1/cGMP signals are highly confined to T-tubular microdomains by local pools of phosphodiesterase 2. This provides a previously unrecognised molecular basis for clearly distinct functional effects engaged by different cGMP producing membrane receptors.

[1] Institute of Experimental Cardiovascular Research, University Medical Center Hamburg-Eppendorf, Martnistr. 52, D-20246 Hamburg, Germany. [2] DZHK (German Center for Cardiovascular Research), partner site Hamburg/Kiel/Lübeck, Martnistr. 52, D-20246 Hamburg, Germany. [3] Clinic of Cardiology and Pulmonology, University Medical Center Göttingen, Robert-Koch-Str. 40, D-37075 Göttingen, Germany. [4] Department of Chemistry, Lund University, Naturvetarvägen 14, SE-221 00 Lund, Sweden. [5] Interfaculty Institute of Biochemistry, University of Tübingen, Hoppe-Seyler-Straße 4, D-72076 Tübingen, Germany. [6] Myocardial Function, National Heart and Lung Institute, ICTEM, Hammersmith Hospital, Imperial College London, Du Cane Road, W12 0NN London, UK. These authors contributed equally: Hariharan Subramanian, Alexander Froese. Correspondence and requests for materials should be addressed to J.G. (email: j.gorelik@imperial.ac.uk) or to V.O.N. (email: v.nikolaev@uke.de)

Elevation of cGMP in adult mouse ventricular myocytes (VMs) by nitric oxide donors, some phosphodiesterase (PDE) blockers, NPs, and inhibitors of NP degradation has been shown to protect the heart from pathological remodelling[1–3]. Correspondingly, pharmacological inhibitors of neprilysin, a peptidase that degrades NPs, have been recently introduced into clinic and are under multiple trials to better understand their action in patients suffering from heart failure with reduced and preserved cardiac contractility or ejection fraction[4]. However, these new and promising drugs increase plasma levels of at least three different NPs, which show major functional differences in terms of their effects on cardiac contractility.

In the heart, atrial natriuretic peptide (ANP), brain natriuretic peptide (BNP) and C-type natriuretic peptide (CNP) increase cardiomyocyte cGMP levels by acting on the membrane guanylyl cyclases natriuretic peptide receptor 1 (NPR1, also known as NPR-A or GC-A) and natriuretic peptide receptor 2 (NPR2, also known as NPR-B or GC-B)[5,6]. NPR1 binds ANP and BNP, while NPR2 binds CNP. ANP and BNP, which are both produced by stretched atria or by diseased VMs[5,6], activate NPR1 and can counteract pathological cardiac hypertrophy[7,8]. However, they show no major effects on contractility[9–12], apart from an occasionally observed slight negative inotropic response[13,14]. In sharp contrast, CNP produced by cardiac fibroblasts and endothelial cells, via its specific receptor NPR2 can markedly affect VM contractility, decreasing the force and improving relaxation, termed as negative inotropic and positive lusitropic effects, respectively[9,11,15]. While both NPR1 and NPR2 produce roughly comparable total amounts of cGMP in human[16] and mouse[17] myocardium, it is unclear why they trigger such markedly different functional effects.

Several lines of evidence suggest that cGMP signals engaged by these two receptors should be differentially compartmentalised[5,6,11,15]. However, due to the lack of powerful real-time imaging techniques and poor sensitivity of available antibodies to detect endogenously expressed receptors in VMs, their exact membrane localisation, and spatial distribution of the produced pools of cGMP are not known. Therefore, we sought to precisely determine submembrane location of functional NPR1 and NPR2 and to understand molecular mechanisms of cGMP compartmentation leading to differences in functional responses to ANP and CNP. Here, we employ the SICM approach combined with FRET, a strategy that we originally developed and successfully used to localise G-protein coupled cAMP stimulating β-adrenoceptors[18]. We show that NPR2 is rather evenly distributed across VM membrane and produces far-reaching, diffusible cGMP signals, whereas NPR1 is exclusively found in T-tubules where it creates a microdomain with restricted cGMP diffusion locally confined by PDE2.

## Results

**SICM/FRET technique for NP receptor localisation.** To localise NP receptors on cell membrane, we scanned VMs freshly isolated from transgenic mice expressing the highly sensitive cytosolic cGMP biosensor[19] red cGES-DE5 and stimulated them with ANP and CNP locally applied onto clearly defined membrane structures from a precisely positioned scanning nanopipette. Using a small pipette diameter (~40–50 nm, ~100 MΩ resistance), we could specifically apply NPs into single T-tubule openings or the outer membrane areas located between the Z-lines, the so-called cell crests (Supplementary Fig. 1). The peptide concentration at the surface was estimated by loading the pipette with 100 μM of the fluorescent dye fluorescein and monitoring the fluorescence at the cell surface before and after delivery with different concentrations of fluorescein in the bath solution. These

measurements showed that the concentration at the surface is ~1/100[th] of the pipette concentration (Supplementary Fig. 2). Finite element simulations using the programme COMSOL Multiphysics were next performed to estimate the concentration profiles on the surface[20]. The simulations showed that when applying from a pipette filled with 100 μM ANP or CNP under bath perfusion, a maximum concentration of ~700 nM can be observed at the membrane with a relatively steep gradient which allows for a ~5–10-time concentration drop at a distance of 2 μm from the activated spot (Supplementary Fig. 3).

**Differential submembrane localisation of NPR1 and NPR2.** When NPR2 was stimulated with CNP, a cGMP signal originating from both T-tubules and cell crests could be detected, suggesting that this receptor is uniformly distributed across various membrane locations (Fig. 1a–c). In sharp contrast, NPR1 stimulation led to cGMP signals only when ANP was applied into T-tubules but not to cell crests, suggesting that this receptor is localised exclusively in T-tubular membranes (Fig. 1e–g). ANP responses originating from T-tubules were generally lower than CNP signals, which might be related to differences in cGMP compartmentation or receptor expression levels. To make sure that these responses reflect the true receptor localisation and are not confounded by different submembrane distributions of the cGMP degrading PDEs, we repeated the same experiment in the presence of the unselective PDE inhibitor 3-isobutyl-1-methylxanthine (IBMX). Even in this case, the ANP response was not detectable at cell crests and remained strictly localised to T-tubules (Fig. 2a, b). To confirm the selectivity of our approach for T-tubules, we performed acute detubulation experiments using dimethylformamide which causes osmotic loss of cell surface connections to the T-tubular system. Detubulation led to blunted ANP responses measured by FRET imaging under global ligand application (Supplementary Fig. 4a–c) or by SICM/FRET (Supplementary Fig. 4d–f). To ascertain that at the locally applied NP concentrations used for SICM/FRET, cGMP signals were specific for the studied receptors, we performed SICM/FRET measurements in CMs isolated from NPR2 knockout mice[21]. In this case, the FRET response to CNP was completely abolished, while the ANP response originating from T-tubules remained fully intact, suggesting high receptor specificity of the applied stimulation protocols (Fig. 2c, d). Since these global NPR2 knockout mice are dwarfs, we have measured the size of single VMs and found it not to be altered as compared to wild-type mice (Supplementary Fig. 5).

To study concentration-response dependencies of SICM/FRET responses to CNP and ANP, we have lowered the applied NP concentrations by ~3.3 and 10 fold. In this case, CNP still induced comparable responses when applied to T-tubules and crests, while ANP responses remained confined to T-tubules (Fig. 3a, b). When compared to concentration-response dependences measured under global NP stimulation by FRET (Fig. 3c, d), around 10 times higher NP concentrations were needed to induce half-maximal response when VMs were stimulated locally in SICM/FRET experiments. This is probably due to local nature of ligand application in these experiments when only a small fraction of the total cellular NP receptor pool gets activated at a time.

**Spatio-temporal profiles of NPR1 and NPR2 induced cGMP.** Having determined the differential localisation of NPR1 and NPR2 at various VM membrane structures, we next analysed the spatio-temporal properties of their respective cGMP signals by looking at changes of FRET at various distances from single stimulated T-tubules. Strikingly, while NPR2 stimulation by CNP induced clearly far-reaching cGMP signals which diffused across

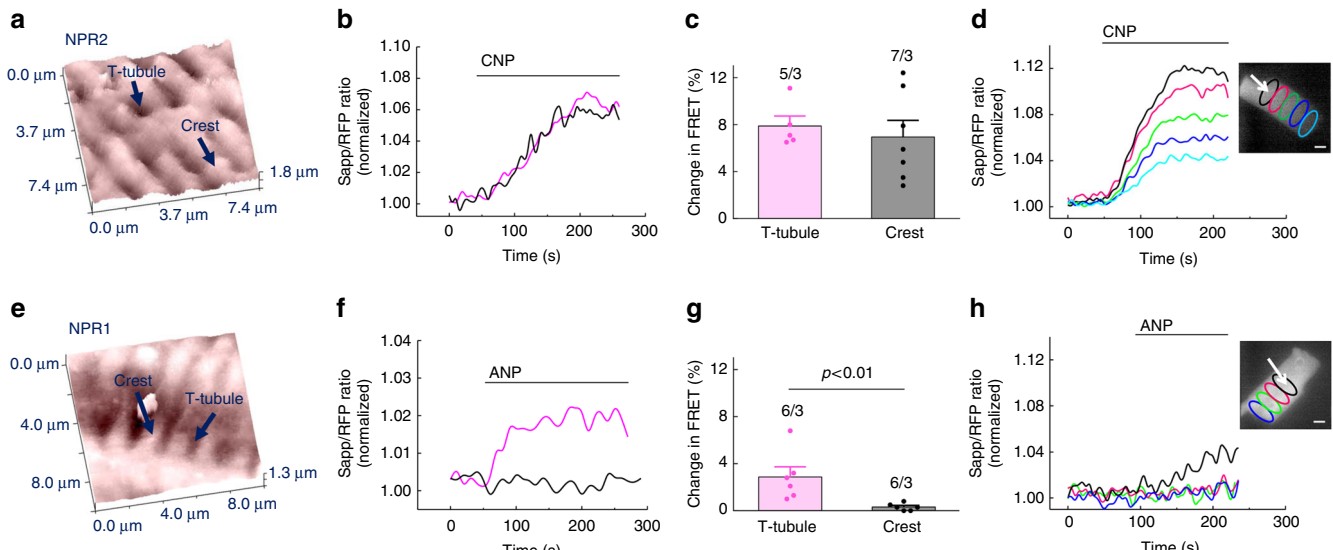

**Fig. 1** Spatial distribution of NPR1 and NPR2 induced cGMP signals in cardiomyocytes. **a**, **b** Representative SICM image and corresponding FRET ratio traces recorded from whole single mouse ventricular cardiomyocytes expressing the cytosolic cGMP biosensor red cGES-DE5 after local NPR2 stimulation on the crest of the cell and in the T-tubule. The cell was superfused with buffer A, and NPR2 was locally stimulated from the scanning nanopipette filled with CNP (100 μM) by applying pressure (276 kPa). Estimated peptide concentration at the membrane was 700 nM (see Supplementary Fig. 3). **c** Quantification of the cGMP-FRET responses to CNP. **d** Spatio-temporal pattern of FRET response measured in different regions of the cell after CNP application to a single T-tubule, showing far-reaching cGMP signals. Scale bar, 10 μm. **e**, **f** Representative SICM image and corresponding FRET ratio traces after local NPR1 stimulation from the scanning nanopipette filled with 100 μM ANP. Quantification is in **g**. All data plotted as means ± s.e.m. Number of cells/mice is above the bars. Difference in **c** is not significant; in **g**, $P < 0.01$ by mixed ANOVA followed by Wald $\chi^2$-test. **h** Spatio-temporal pattern of FRET signals after ANP application to a single T-tubule shows a highly locally confined cGMP response. Scale bar, 10 μm. Quantification of experiments shown in **d** and **h** is in Fig. 6b

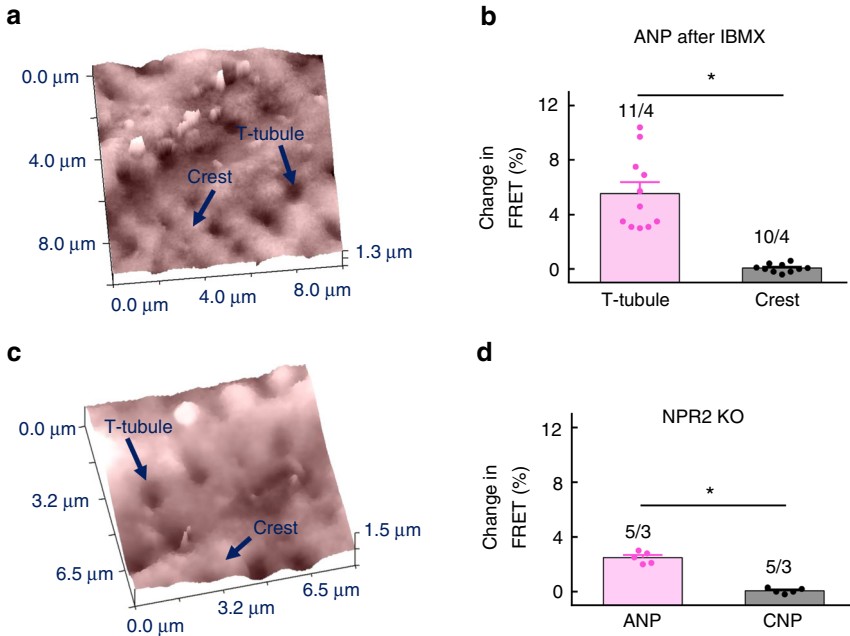

**Fig. 2** Control SICM/FRET experiments. Recordings were performed in presence of 100 μM IBMX (unselective PDE inhibitor, preincubated for 3–5 min in the bath solution) (**a**, **b**) or in NPR2 knockout myocytes (**c**, **d**) as described in Fig. 1. Representative scans and data quantifications are shown. These data rule out the possibility that local PDEs might affect the specific NPR1 localisation measured by SICM/FRET in Fig. 1e–g and confirm the specificity of the applied stimulation protocol for the studied receptors/ligands. Data represent mean ± s.e.m. values of the indicated number of cells/mice. *$P < 0.01$ (Kruskal–Wallis test)

the entire cell, NPR1/cGMP signals were strictly locally confined and detectable only very close to the site of stimulation (Fig. 1d, h). This can potentially explain why ANP affects only local T-tubular pools of cGMP effector proteins involved in the

regulation of the L-type calcium channels[22,23] localised in this microdomain[24] to induce minor negative inotropic responses, whereas CNP-stimulated cGMP leads to phosphorylation of phospholamban (PLN) and troponin I responsible to positive

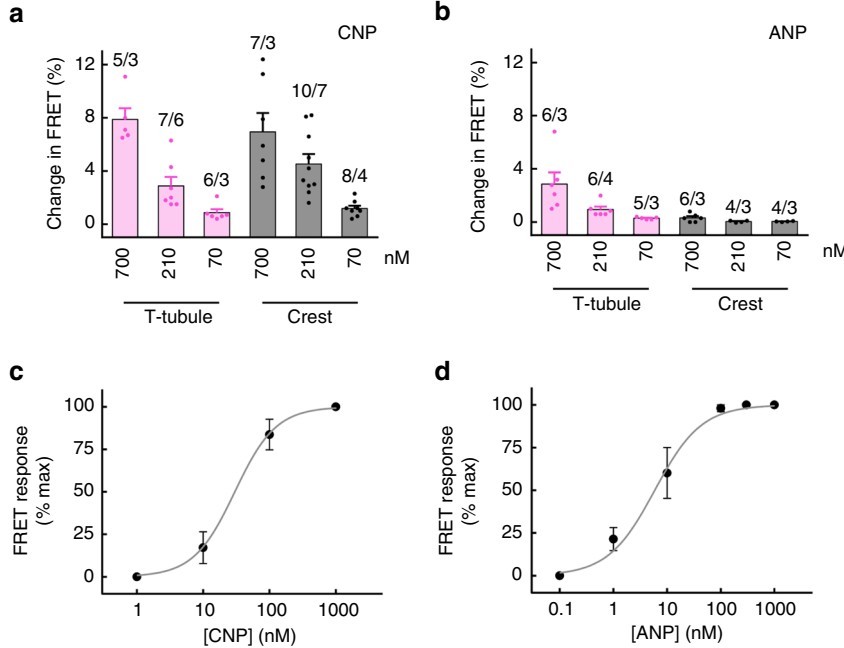

**Fig. 3** Concentration-response dependences of CNP and ANP signals. Responses to different ANP and CNP concentrations were recorded either using SICM/FRET (**a**, **b**) as described in Fig. 1, or by simple FRET imaging with global ligand application into the bath solution (**c**, **d**). In **a**, **b**, calculated peptide concentrations at the membrane are shown. 700, 210, and 70 nM were obtained by filling SICM pipetted with 100 μM, 30 μM, and 10 μM of CNP or ANP, respectively. Data for 700 nM are reproduced from Fig. 1 to facilitate the comparison. Data represent mean ± s.e.m. In **a**, **b**, numbers of cells/mice are shown above the bars. In **d**, 9–12 cells from three mice each were analysed for each data point. In **c**, data are reproduced from our previous study[19] for comparison

lusitropic and negative inotropic responses, respectively[15,25], and located much deeper inside the cell.

**Mechanisms of differential receptor localisation**. To further investigate which mechanisms might be responsible for differential localisation of both receptors, we hypothesised that interaction of NPR1 with caveolin-rich membrane domains or lipid rafts present in T-tubuli might play a role. Indeed, when VMs were incubated with the cholesterol depleting agent methyl-β-cyclodextrin (MβCD), NPR1/cGMP signals redistributed from T-tubules to cell crests (Supplementary Fig. 6a–c). In contrast, the amplitude and localisation of the NPR2/cGMP were not affected by methyl-β-cyclodextrin treatment (Supplementary Fig. 6d, e). To test which structural features of NPR1 might be responsible for its interaction with caveolin-rich membrane domains, we developed receptor constructs in which NPR1 and NPR2 were fused to enhanced yellow fluorescent protein (EYFP) to study receptor mobility by fluorescence recovery after photobleaching (FRAP) in transfected HEK293A cells. Despite overall high similarity, NPR1 and NPR2 protein sequences differ considerably in the transmembrane domain (TMD), so that we decided to clone chimeric receptors with swapped TMDs (Supplementary Fig. 7). FRAP microscopy revealed that NPR1-EYFP but not NPR2-EYFP mobility was, as expected, sensitive to MβCD treatment suggesting its localisation in lipid rafts or caveolin-rich domains (Fig. 4a, c, d, f). A decrease in mobility after MβCD is compatible with the behaviour of proteins located in these membrane compartments[26,27]. Interestingly, exchange of NPR1 TMD with that of NPR2 completely abolished MβCD sensitivity (Fig. 4b, c), suggesting that TMD is important for localisation of NPR1 to caveolin-rich domains. Conversely, exchange of NPR2 TMD with the of NPR1 has regained MβCD sensitivity but led to a decrease in mobility (Fig. 4e, f). Different direction of MβCD effect might indicate that isolated TMD could be just one of

several structural features required for proper receptor localisation and mobility. On the other hand, membrane microdomain structures of HEK293A cells differ from those of myocytes which might affect the behaviour of such constructs.

**Imaging local cGMP and its regulation at the membrane**. To uncover the molecular mechanisms responsible for the strict compartmentation of ANP/cGMP signals at the T-tubular membranes, we generated a FRET-based cGMP biosensor pmDE5 targeted to caveolin-rich membrane domains including T-tubules and expressed it in CMs of transgenic mice (Fig. 5a, b). This targeted version of red cGES-DE5 is particularly well suited to analyse membrane-associated compartmentation mechanisms, as previously established for cAMP microdomains[10]. The presence of the sensor did not affect proper heart function and could report local cGMP increases in response to NPs and PDE inhibitors (Supplementary Fig. 8). Measurements using this plasma membrane-targeted biosensor revelled that at basal state or after CNP stimulation, PDE2, PDE3 and PDE5 inhibitors showed relatively small responses, with PDE3 producing a slightly stronger effect (Fig. 5c, d). However, strikingly, after ANP stimulation, the PDE2 inhibitor BAY 60-7550 led to a remarkable increase of cGMP and could unmask local ANP responses (Fig. 5e, f), while PDE3 and PDE5 inhibitors had no effect. This suggests that PDE2 plays the major role in compartmentalising ANP/NPR1/cGMP signals at the T-tubular membrane.

**Role of PDE2 in compartmentalisation of NPR1-derived cGMP**. To further investigate this hypothesis, we tested whether PDE2 inhibition could transform locally confined ANP/cGMP signals into far-reaching ones. We pretreated red cGES-DE5 expressing VMs with BAY 60-7550 and applied ANP into single T-tubular openings using SICM/FRET, similar to the experiments performed in Fig. 1e–h. In this case, PDE2 inhibition now led to

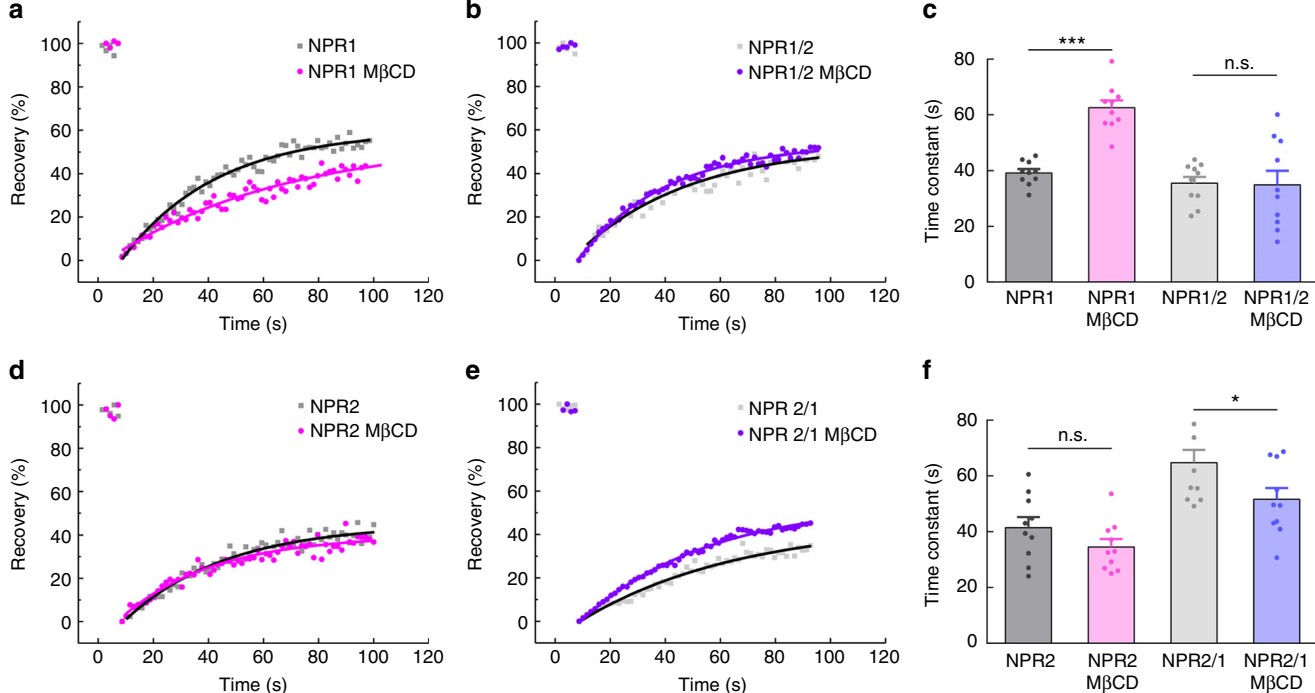

**Fig. 4** Fluorescence recovery after photobleaching (FRAP) analysis for various NPR1-EYFP and NPR2-EYFP constructs. HEK293A cells were transfected with receptor constructs described in Supplementary Figure 7 and imaged 24–30 h later with and without MβCD treatment (1 mM for 1 h at 37 °C). Fluorescence was measured and bleached in a rectangular cell membrane region of interest using 488 nm laser and confocal microscope as described in Methods. Representative fluorescence recovery curves (**a**, **b**, **d**, **e**) and their time constant analysis after monoexponential fit (**c**, **f**) are shown. **a** NPR1-EYFP receptor mobility is sensitive to MβCD treatment. **b** This effect is abolished when TM domain of NPR1 is replaced by that of NPR2 in the NPR1/2 construct. **d** NPR2-EYFP receptor mobility is not sensitive to MβCD treatment. **e** However, it is changed by MβCD in the NPR2/1 construct which contained TM domain derived from NPR1. The opposite direction of MβCD effect (as compared to wild-type NPR1, compare **e** and **a**) might indicate that TM domain is crucial but not the only factor which affects NPR1 mobility. Data represent mean ± s.e.m. $n = 10$ cells each. * $p < 0.05$; *** $p < 0.001$; n.s., not significant by one-way ANOVA

far-reaching ANP/cGMP signals (Fig. 6a, b), corroborating the central role of PDE2 in compartmentation of ANP/NPR1 signalling. In line with these data, immunoblot analysis of PLN phosphorylation in working hearts showed virtually no effect of ANP alone, as expected based on previous reports[15,25]. However, while the PDE2 inhibitor BAY 60-7550 alone showed no significant increase, ANP led to a robust increase of phospholamban phosphorylation in presence of this inhibitor (Fig. 6c, Supplementary Fig. 9). Although PDE2 does not seem to directly interact with NPR1 based on co-immunoprecipitation experiments (Supplementary Figs. 10, 11), ANP-induced PLN phosphorylation could be unmasked by incubating VMs with MβCD (Supplementary Figs. 12, 13), suggesting that proper localisation of NPR1 to caveolin-rich membrane domains including T-tubuli is important for compartmentation of its cGMP signals.

## Discussion

Based on our findings, the following model of NP/cGMP signalling can be proposed. NPR2, which is located in various regions of VM sarcolemma, generates cGMP signals that are not highly controlled by PDEs and are capable of diffusing throughout the entire cell. This enables the cGMP-dependent protein kinase (cGK) to phosphorylate its substrates located deep inside the cell. Among these substrates, PLN and troponin I are the main mediators of the well-established positive lusitropic and negative inotropic effects of CNP/NPR2/cGMP[15,25] (Fig. 6d). It is well documented that CNP stimulation of VMs leads to PLN phosphorylation, which is completely abolished in cGK knockout cells[25]. In addition, NP-stimulated cGMP can participate in the so-called positive cGMP-to-cAMP cross-talk, inhibiting PDE3

and increasing cAMP levels, which can further enhance electro-mechanical coupling, especially under concomitant beta-adrenergic stimulation[19,28]. In sharp contrast, ANP via NPR1 located in the T-tubular membranes generates highly compartmentalised cGMP signals, which do not phosphorylate PLN and can act only in the vicinity of this receptor on the cGK substrates or PDEs located in this microdomain, such as L-type calcium channels[24] and PDE2, presumably responsible for the mild negative inotropic effect of ANP[23] (Fig. 6d). Thus, the differential submembrane localisation of both particulate guanylyl cyclases determines the spatial profile of intracellular cGMP response and hence their physiological effects and functional roles in the heart.

Interestingly, ANP/NPR1 pathway has been shown to functionally interact with transient receptor potential subfamily C (TRPC) channels to prevent cardiac hypertrophy[29] via cGKI mediated phosphorylation, which inhibits channel activity[30]. It is well documented that TPRC channels are located in T-tubules of VMs[31,32] and can even form a stable molecular complex with NPR1 (ref. [33]). So it is tempting to speculate that localisation of NPR1 in T-tubules promotes its molecular and functional association with TPRC channels to mediate antihypertrophic effects of ANP and BNP.

Using a specifically targeted cGMP biosensor for this microdomain, we could show that ANP/NPR1/cGMP signal compartmentation at the T-tubular membranes is possible due to the local action of PDE2. This PDE has previously been implicated in the local submembrane control of ANP/NPR1/cGMP signals measured using more evenly localised cyclic nucleotide gated channels as sensors for subsarcolemmal cGMP[34] and also in the regulation of the local cGMP/cAMP cross-talk close to β₁- and

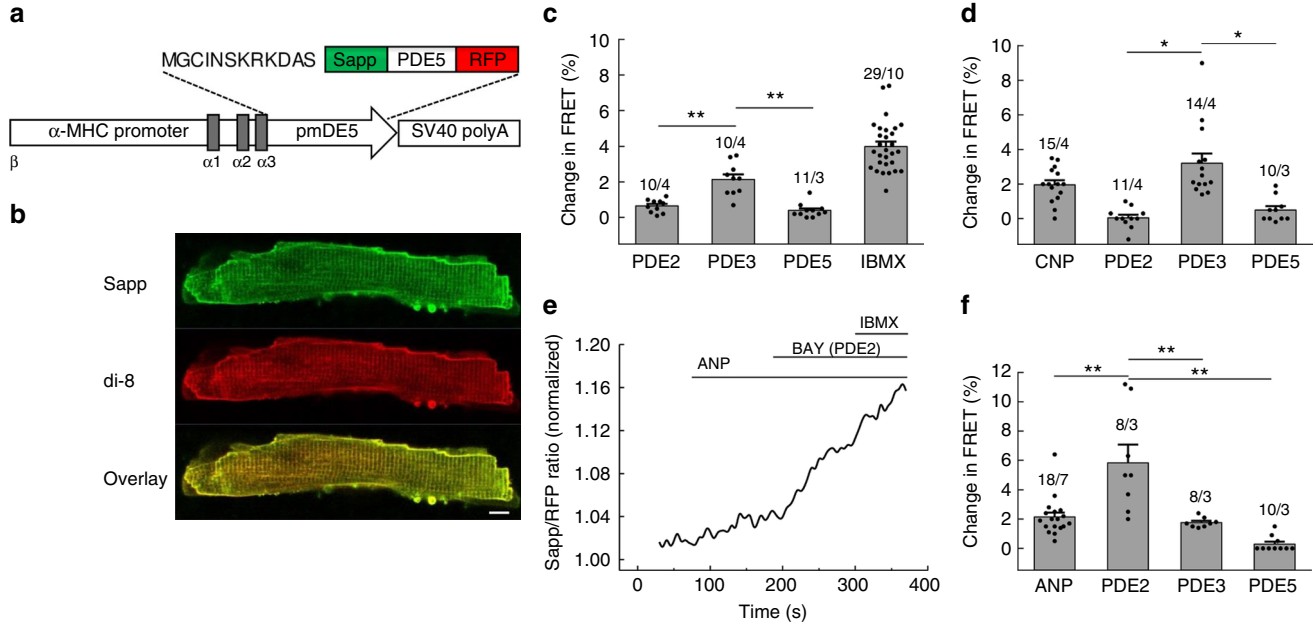

**Fig. 5** T-tubular ANP/NPR1/cGMP responses are under tight local control by PDE2. **a** Schematic representation of a genetic construct used to generate pmDE5 transgenic mice expressing this targeted biosensor in adult VMs under the control of the α-myosin heavy chain (α-MHC) promoter. **b** Live-cell confocal imaging of isolated pmDE5 myocytes stained with di-8-ANEPPS (di-8) confirms the proper localisation of the sensor to caveolin-rich membrane domains. Representative images, $n = 10$. Scale bar, 10 μm. **c, d, f** Quantification of FRET responses recorded upon application of the selective PDE inhibitors BAY 60-7550 (PDE2, 100 nM), cilostamide (PDE3, 10 μM) and tadalafil (PDE5, 100 nM) either alone at basal state (**c**) or after 100 nM CNP (**d**) and 100 nM ANP (**f**) prestimulation. Data are means ± s.e.m. Number of cells/mice is above the bars. *$P < 0.01$, **$P < 0.01$ by mixed ANOVA followed by Wald $\chi^2$-test. In **c**, **d** and **f**, Sapp/RFP FRET ratio before addition of PDE inhibitors (at basal state of after NP stimulation) was normalised on 1 to calculate % changes in FRET ratio and to facilitate the comparison between various conditions. **e** Representative FRET trace from pmDE5 myocyte subsequently stimulated with 100 nM ANP, 100 nM BAY 60-7550 and 100 μM IBMX, showing a marked response to the PDE2 inhibitor after ANP prestimulation

β3-adrenoceptors[10,35]. However, it was unclear how this local action of PDE2 can modulate spatial gradients of cGMP inside the cell. The PDE2-dependent mechanism uncovered in our study is crucial for restricting cGMP diffusion away from the T-tubular membranes into the cell cytosol and for preventing ANP/cGMP effects on sarcoplasmic reticulum and on contractile proteins. The exact mechanisms responsible for subcellular PDE2 localisation in VMs are not well understood, although one study performed in neurons, showed that dual myristoilation at the N-terminus of PDE2A3 is responsible for its membrane targeting[36].

In addition to explaining the distinct cGMP-dependent functional effects of NPR1 and NPR2 in cardiomyocytes, our findings provide a mechanism of potential therapeutic relevance. While both ANP/BNP and CNP seem to counteract pathological cardiac remodelling occurring in heart failure, their differential effects on contractility should be considered when treating patients with reduced vs. preserved ejection fraction, the latter having diastolic dysfunction, which might be better addressed by CNP than ANP/BNP. Specifically preventing degradation of individual NPs, combination of neprilysin inhibitors with particular peptidase-resistant designer NPs[37] or even with pharmacological tools modulating PDE2 activity[38,39] could be considered as new therapeutic options for cardiac disease.

## Methods

**Chemicals.** Mouse/rabbit/rat atrial natriuretic peptide (1–28) was from Bachem, C-type natriuretic peptide was purchased from Merck. IBMX was from Applichem, BAY 60–7550 from Santa Cruz. All other chemical were from Sigma-Aldrich.

**Cardiomyocyte isolation.** All mouse work was done according to international animal welfare guidelines and approved by national authorities LAVES Niedersachsen (Ref. No. 33.14–42502–04-10/0200) and BGV Hamburg (ORG 741). Adult mouse ventricular cardiomyocytes were isolated using the Langendorff perfusion and enzymatic digestion as previously desribed[40]. Briefly, red cGES-DE5

or pmDE5 transgenic mice were euthanised, the hearts were quickly excised, cannulated and perfused at 37 °C with calcium-free perfusion buffer (in mM: NaCl 113, KCl 4.7, KH$_2$PO$_4$ 0.6, Na$_2$HPO$_4$x2H$_2$O 0.6, MgSO$_4$x7H$_2$O 1.2, NaHCO$_3$ 12, KHCO$_3$ 10, HEPES 10, Taurine 30, 2,3-butanedione-monoxime 10, glucose 5.5, pH 7.4) for 3 min followed by 30 ml digestion buffer containing liberase DH (0.04 mg/ml, Roche), trypsin (0.025%, Gibco) and 12.5 μM calcium chloride. After stopping the digestion with 10% foetal calf serum and recalcification, the cells were plated on laminin coated glass-bottomed Fluorodishes (World Precisicon Instruments) and incubated at 37 °C and 5% CO$_2$ until use. In the case of NPR2 knockout mice (B6;129S7-Npr2$^{tm2.1(CreERT2)Fgr}$)[21], the cells were transduced with red cGES-DE5 adenovirus for 40–48 h prior to SICM/FRET measurements.

**SICM/FRET.** SICM is a scanning probe microscopy technique based on measuring ion current flowing through a tip of a glass nanopipette (~100 MΩ resistance, corresponding to 40–50 nm inner diameter which determines the spatial resolution of the scan). The pipettes were filled with buffer A (144 mM NaCl, 5.4 mM KCl, 1 mM MgCl$_2$, 1 mM CaCl$_2$ and 10 mM HEPES, pH = 7.3) containing NPs. The lower vertical (z-axis) position of the pipette tip was kept constant relative to the cell surface[41]. In this study, we used the hopping probe ion conductance microscopy[42] and the ICnanoS scanning system (Ionscope, Melbourn, UK) or similar custom designed SICM system (ICAPPIC, London, UK) positioned on the Nikon Eclipse Ti inverted fluorescence microscope. During application of receptor ligands, we minimised the hopping amplitude and superfused the cell with the buffer A to ensure a strictly local receptor stimulation (<500 nm) achieved by pressure application (276 kPa, 40 psi) to the pipette as previously described[18]. The cells were kept in buffer A with or without 100 nM of the PDE2 inhibitor BAY 60–7550 or 100 μM of the unselective PDE inhibitor IBMX.

Simultaneous FRET imaging of cGMP in stimulated CMs (as well as simple FRET recordings in pmDE5 VMs) was performed using a homemade imaging system, consisting of a 400 nm pE-1000 light emitting diode (CoolLED) to excite the biosensor, the microscope (as above) and a DualView beam splitter (Photometrics) equipped with 565dcxr dichroic mirror, D520/30 and D630/50 emission filters. Fluorescence in individual channels was recorded by the ORCA-03G camera (Hamamatsu) using the open source MicroManager 1.4 software[43]. FRET ratios were corrected for the bleedthrough of the T-Sapphire into the Dimer2 channel and analysed using Origin 8.0 software as described[19,44].

**Biosensor cloning and transgenic mouse generation.** To generate the membrane-targeted cGMP biosensor termed pmDE5, the DNA encoding for the

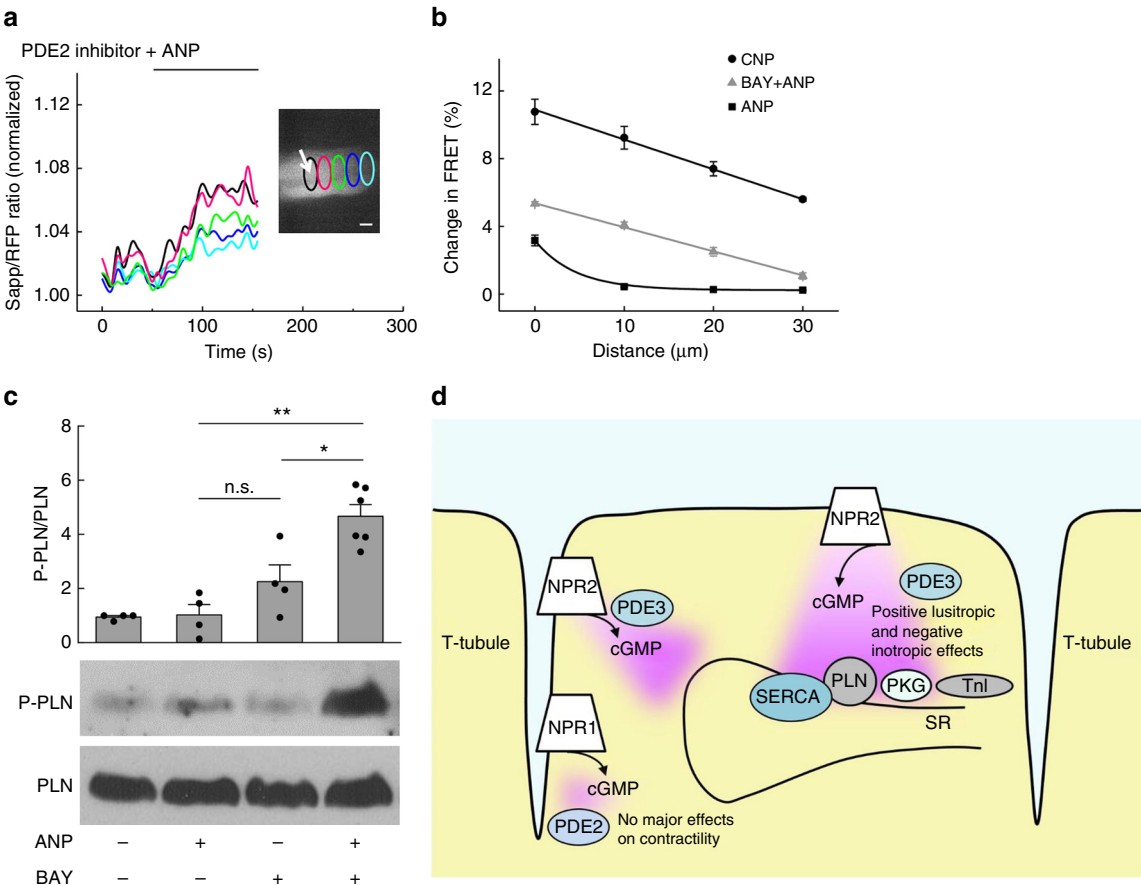

**Fig. 6** Differential subcellular compartmentation of NPR1 and NPR2/cGMP responses. **a** Spatio-temporal pattern of FRET signals after ANP application shows far-reaching cGMP response after pre-incubation with 100 nM of the PDE2 inhibitor BAY 60-7550. Scale bar, 10 µm. Quantification is in **b**. **b** includes also the quantification of experiments shown in Fig. 1d, h. Shown are means ± s.e.m five cells from three mice per condition were analysed. Differences between all three conditions were significant at $p = 0.05$ for each individual distance value using mixed ANOVA followed by Wald $\chi^2$-test. **c** Immunoblot analysis of phospholamban (PLN) phosphorylation on Ser16 in working hearts perfused with 100 nM ANP with or without 100 nM BAY 60-7550. * $P < 0.05$, ** $P < 0.01$, and n.s. not significant by one-way ANOVA. **d** Schematic diagram illustrating the major findings of the study. NPR2 is widely distributed across the membrane, generating far-reaching cGMP responses. This cGMP pool is under mild control by PDE3 and causes phosphorylation of PLN and troponin I (TnI), leading to negative inotropic and positive lusitropic effects. NPR1 is exclusively found in T-tubules where it generates locally confined cGMP signals restricted in this microdomain by local action of PDE2. This explains why there are no major effects of ANP/NPR1 on cardiac contractility. SR sarcoplasmic reticulum, SERCA SR calcium ATPase

N-terminal 10 amino acid peptide MGSINSKRKD from the Lyn kinase was sub-cloned prior to the start codon of the red cGES-DE5 sensor[45] using the KpnI and NheI restriction sites. The full sensor sequence was next subcloned into the previously described vector containing the α-myosin heavy chain promoter and simian virus (SV40) polyadenylation signal[46]. The resulting vector was linearised with SpeI, purified and used for pronuclear injections to generate transgenic mice on the FVB/N1 background as previously described[10,46]. Founder mice and their heterozygote offspring were genotyped by a standard PCR using the primers TGA-CAGACAGATCCCTCCTAT and GGATGCTCAGGTAGTGGTT (see Supplementary Table 1 for full list of primers), resulting in a ~690 b.p. fragment on a gel.

**Confocal microscopy.** Confocal microscopy was performed using Zeiss LSM 710 NLO microscope (Carl Zeiss MicroImaging, Jena, Germany) equipped with a Plan-Apochromat ×63/1.40 oil-immersion objective. For membrane staining, myocytes were incubated with 50 µM of di-8-ANEPPS for 10–15 min[10]. Images were acquired for T-sapphire (Sapp, 405 nm diode laser excitation) and di-8-ANEPPS (488 nm argon ion laser excitation) channels and analysed using ZEN 2010 software (Zeiss).

**FRAP measurements.** Mouse NPR2 cDNAs was amplified using primers AAAAGCTTATGGCACTGCCATCCC (NPR2 forward) and AAA-GAATTCGCAGGAGTCC GGGAGG (NPR2 reverse) and cloned into EYFP-N1 vector via HindIII and EcoRI. NPR1-YFP plasmid has been previously described[33] (see Supplementary Table 1 for full list of primers). Chimeric NPR1 (NPR1/2) having NPR2 transmembrane domain (TMD) and vice versa were developed as

follows. The extracellular domain of NPR1 was amplified using NPR1 forward primer (AAAGCTAGCGCCATGCCGGGCTCC) and a NPR1 TMD-Rev primer (GAAAATTAGGAAACTGGAAACACCAAACATGATGAAGGTGACTCCCG TGCCCAGGGGCCACCTCCAGTGTGGAAAAGTGG). Cytosolic domain of NPR1 was amplified using NPR1 TMD-Forw primer (GTGGCCCTGGGCACGG-GAGTCACCTTCATCATGTTTGGTGT TTCCAGTTTCCTAATTTT CAGGAA-GATGCAGCTGGAA) and NPR1 reverse primer (AAACTCGAGTCCCCTA GTGCTACATCCCCGCT). Then, by overlapping extension PCR, the chimeric cDNA of NPR1/2 was amplified using NPR1 forward and NPR1 reverse primers and the amplified cDNA was cloned into the EYFP-N1 vector. NPR2/1 chimera was made using the same strategy as for NPR1/2. The primers used are NPR2 forward and NPR2 TMD-Rev (GTATATGAAGAAAGACACAATCA-GAAAGCTAATCAGAGAGAGGCTGCCCACCA GGGCCACGATTGCC) for amplifying NPR2 extracellular domain, and a NPR2 TMD-Forw primer (GTGGGCAGCCTCTCTCTGATTAGCTTTCTGATTGTGTCTTTCTTCATATACCGGA AGCTGATGCTGGA) and NPR2 reverse primer for amplifying the cytosolic domain. HEK293A cells (293 A cells from ThermoFischer Scientific, Catalogue Number R70507, Mycoplasma free tested) were grown in Dulbecco's Modified Eagle Medium supplemented with 10% foetal calf serum, L-glutamine and antibiotics at 37 °C, 5% $CO_2$, plated on round glass coverslides and transfected 24 h after plating with the wild-type or chimeric plasmids using Lipofectamine 2000 (Life Technologies) following manufacturer's protocol. 24 hours of post-transfection, the cells were treated with 1 mM methyl-β-cyclodextrin (MβCD) or vehicle for 1 h before measurements. The cells were incubated in Buffer A and mounted on the confocal microscope as above. A small rectangular area of interest (~2 µm²) on the cell membrane was selectively bleached with 100% laser power (488 nm) and the fluorescent intensity within the selected region was measured every 1.45 s with

~1% laser power before and after bleaching. The fluorescence recovery curves were plotted with relative fluorescence intensity in the bleached area against time. The fluorescence recovery half time was calculated for each curve with mono-exponential fit using Origin software.

**Echocardiography**. Echocardiography of wild-type and transgenic pmDE5 mice was performed at the age of 6 months using the Vevo 2100 system (VisualSonics) equipped with a 30-Hz transducer (MS-400 MicroScan Transducer) as described[47].

**Computer based simulations**. Finite elements simulations were made using the programme COMSOL Multiphysics 5.3. The simulations were similar to those described in Babakinejad et al.[20], but did also include the effect of perfusion. The parameter values given in Supplementary Table 2 were assumed and the simulation geometry were changed from a ¼ pipette geometry in Babakinejad et al. to a ½ pipette geometry, with the rectangular, bounding domain extending between $-30$ $\mu m < \times < 40 \, \mu m$, $0 < y < 30 \, \mu m$ and $0 < z < 30 \, \mu m$, with the pipette axis coinciding with the $z$-axis. The effective pressure being applied over the pipette was estimated from the delivery of fluorescein summarised in Supplementary Figure 3. The total flow out of the pipette, $Q_{tot}$, can be determined from the expression[20]:

$$Q_{tot} = -4\pi Dh \ln(1 - c_{max}/2c_0) \tag{1}$$

where $h$ is the pipette-surface distance, $D$ the diffusivity of the delivered molecule and $c_{max}/c_0$ the ratio between the maximum concentration on the surface and the concentration in the bulk of the pipette. From delivery of the fluorescent dye fluorescein, under similar conditions to those used when delivering ANP and CNP, it was found that $c_{max}/c_0 \approx 0.01$ which together with $D = 4.25 \times 10^{-10} \, m^2/s$ for fluorescein[48] and $h = 1 \, \mu m$ gives $Q_{tot} = 2.7 \times 10^{-17} \, m^3/s$. With the assumed pipette dimensions in Supplementary Table 2 and using Eq. 15 in Babakinejad et al.[20] this corresponds to an effective pressure of 6.8 kPa over the pipette.

Two creeping flow simulations were performed to determine (i) the flow out of the pipette due to the applied pipette pressure and (ii) the perfusion velocity around the pipette, which is assumed to be in the $x$-direction. The boundary conditions for these simulations are summarised in Supplementary Table 3. The velocity inside the pipette was set to zero in the perfusion simulations. The obtained velocities from the two simulations were added and used as the convective velocity when solving for the concentration profile as described previously in Babakinejad et al.[20] with the concentration set to zero at the outer boundaries of the simulation geometry.

**Immunoblot analysis of phospholamban phosphorylation**. Whole beating hearts were cannulated and perfused with Krebs–Henseleit buffer (KH; in mM: 118 NaCl, 4.7 KCl, 1.2 $KH_2PO_4$, 1.25 $MgSO_4$, 24 $NaHCO_3$, 1.25 $CaCl_2$ and 11.1 glucose; constantly oxygenated with 95% $O_2$ and 5% $CO_2$) at 37 °C. Fifteen minutes after equilibration, hearts were perfused with 100 nM ANP and/or 100 nM BAY 60-7550 in KH buffer for 10 min. After treatment, the left ventricle was dissected and shock-frozen. Frozen hearts were homogenised in a buffer containing 10 mM HEPES, 300 mM sucrose, 150 mM NaCl, 1% triton ×100, protease and phosphatase inhibitor tablets (both from Roche). Alternatively, isolated cardiomyocytes were incubated with 0.5 mM MβCD for 45 min at 37 °C, and then treated with 100 nM ANP for 10 min before homogenisation. A volume of 20 µg of protein homogenates were run on SDS-PAGE and transferred to nitrocellulose membranes. pPLN[Ser16] and total PLN were detected using specific antibodies purchased from Badrilla (catalogues numbers A010-12 and A010-14, respectively, both used at the dilution 1:5000). Blots were quantified using ImageJ software. Uncropped scans of all representative blots shown in the manuscript are provided in supplementary information (Supplementary Figs. 9, 11 and 13).

**Co-immunoprecipitation analysis**. HEK293A cells were transfected with mouse PDE2A3 and/or FLAG-tagged NPR1 (kind gift from Michaela Kuhn) plasmids, using Lipofectamine 2000 (Life Technologies) following manufacturer's protocol. After 48 hours of post-transfection, the cells were lysed with 20 mM Tris, 1% triton and 120 mM NaCl, 2 mM EDTA, 2 mM EGTA, and protease and phosphatase inhibitors (both from Roche). After removing debris by centrifugation, cell lysate was incubated with either mouse anti-FLAG antibody (Sigma, catalogue number M1804) or goat anti-PDE2A3 antibody (Santa Cruz, catalogue number sc17228) at 4 °C overnight. Then, using Protein A/G agarose (Santa Cruz) the immunoprecipitated complex was recovered and eluted with Laemmli buffer. The eluted products were subjected to immunoblotting (antibody dilutions 1:1000).

**Statistics**. All data shown in bar graphs are presented as mean ± s.e.m. of $n$ independent experiments. In FRET and SICM/FRET experiments, n represents number of measured cells isolated from N mice, numbers of $n/N$ analysed for each condition are indicated above the individual bars. Sample sizes were 3–6 mice and 5–13 measured cells. Statistical analyses were performed with the OriginLab 8.0 and R 3.0.3 software. Normal distribution was tested by the Kolmogorov–Smirnov test, and differences between the groups were analysed using mixed ANOVA followed by Wald $\chi^2$-test, or Kruskal–Wallis ANOVA, as appropriate. Echo-cardiographic, FRAP and immunoblot data were analysed by one-way ANOVA.

Statistical tests were apply only if variance was similar between the groups. No sample exclusion was performed.

**Data availability**. Data supporting the findings of this manuscript are available from the corresponding authors upon reasonable request.

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

## Acknowledgements

We thank the core transgenic unit of the Max-Planck-Institute for Experimental Medicine for transgenic mouse generation, Sabrina Wolborn for echocardiography, Karina Schlosser for technical assistance, and Laurinda Jaffe for critically reading the manuscript. This work was supported by the Deutsche Forschungsgemeinschaft research group FOR2060 (to V.O.N and H. Schmidt), British Heart Foundation (grant RG/17/13/33173 to J.G.) and the Gertraud und Heinz Rose-Stiftung (grant to V.O.N.).

## Author contributions

V.O.N. and J.G. designed the project. A.F. performed SICM/FRET studies. H.S. performed immunoblots and FRET imaging experiments in pmDE5 CMs. P.J. performed computer simulations. H.S. provided NPR2 knockout mice. All authors wrote and edited the manuscript.

## Additional information

**Competing interests:** The authors declare no competing interests.

