## [Peer Review File · Nature Communications]

Reviewers' comments:

Reviewer #1 (Remarks to the Author):

Subramanian and co-authors have submitted a manuscript in which they have investigated the subcellular localization of cGMP signaling downstream of natriuretic peptide receptors A and B (NPR-A, NPR-B). This was accomplished using ANP and CNP as activators of NPR-A and NPR-B, respectively, along with SICM and FRET based experimental approaches in isolated mouse ventricular myocytes.

The area of investigation is very interesting and important and has potential implications for how natriuretic peptides (NPs) can/should be used therapeutically for heart disease patients. Overall, the manuscript is well written and straightforward. The findings of the study potentially provide some explanation for differential effects of ANP and CNP on cardiac contractility.

While the study is generally well done, I have several comments that the authors should address in order to improve their study and strengthen their conclusions.

1. The use of SICM and FRET combines for a powerful experimental approach. Nevertheless, some clarification would be helpful. For example, the conclusions of the study depend on the ability to precisely deliver compounds to a single t-tubule or to the cell crests. Have the authors tried disrupting the t-tubules to confirm the selectivity of their approach (i.e. can you abolish the effects of ANP on cGMP production by disrupting the T-tubules chemically)?
2. The authors also need to more comprehensively assess the dose dependence of NPs in their experiments. The doses used are unclear. The authors state that the pipette contained 100 μM of ANP or CNP (a very high concentration) and suggest that actual concentrations at the receptors are 'well below 1 μM ' (which is still a concentration that is much higher than that in the circulation). I realize that local concentrations are likely higher than circulating levels, but it would be informative to present data in which the predicted concentrations at the receptors are in the low nanomolar range.
3. If the authors conclusion is that ANP mediated production of cGMP is restricted by PDE2, why does ANP application in the presence of IBMX (which will inhibit PDE2 and other PDEs) produce a similar effect on cGMP signaling as seen in the absence of IBMX (supplemental Fig. 2G vs. Fig 1G). One would expect a larger effect of ANP following PDE inhibition.
4. The authors state that NPR-A and NPR-B produce similar amounts of cGMP, but that the subcellular distribution of the two receptors is different. In fact, the authors data suggest that the amount of cGMP produced in the t-tubule compartment by CNP is much greater than ANP (Figure 1C and 1G). Please comment and, if necessary, revise the manuscript.
5. The authors have used Npr2 knockout mice to confirm that the effects of ANP are mediated by NPR-A. please clarify the phenotype of these Npr2 knockout mice as global Npr2 knockouts are dwarfs, which may impact your study.
6. The findings with M β CD and ANP are interesting. What are the effects of CNP in the presence of this compound? It would also be helpful to present representative data in addition to the summary bar graphs (Supplemental Figure 3).
7. Although inhibition of PDE2 does appear to facilitate farther reaching ANP mediated cGMP signals, they are still substantially smaller than those observed with CNP. Is it possible that other PDEs (or molecules other than PDEs) are also involved? Or is this reflective of your data that seem to suggest CNP produces more cGMP than ANP? Please comment.

8. The authors need to be specific throughout their manuscript that they are using ventricular myocytes. Using 'cardiomyocytes' is not clear enough. This is important because it is conceivable that very different findings would be observed in atrial or sinoatrial node myocytes, which have different cellular morphology and different patterns in the t-tubular network. Indeed, studies have shown, for example, that BNP and CNP have very similar effects on heart rate and ion currents in the sinoatrial node.

9. The authors may be missing an important contribution of PDE3 in the interpretation of their findings. cGMP produced downstream of NPRs can inhibit PDE3 (including at lower concentrations than those required to activate PDE2), which can result in an elevation in cAMP-PKA signaling. The authors show some effects of PDE3 inhibition in their results and this could also impact on, for example, phosphorylation of PLB and other targets. The authors should address this in their discussion.

Minor comments

1. The authors have oversimplified the introduction (page 3) by stating that ANP is an atrial peptide and BNP is a ventricular peptide. In fact, ANP and BNP are both present in atrial granules in the normal heart. Similarly, particularly in heart failure, the ventricles produce BNP and ANP. The authors should clarify their introduction and provide appropriate citations.
2. Please present a comprehensive panel of echocardiography findings (Supplemental Figure 4). The authors should include chamber dimensions and wall thicknesses in systole and diastole along with heart rate etc.
3. The authors should consider moving some/all of the supplemental data into the main paper as there is plenty of room to do so and the results in these figures are important.

Reviewer #2 (Remarks to the Author):

The manuscript titled "Distinct submembrane localisation compartmentalises cardiac NPR1 and NPR2/cGMP signalling by Subramanian et al. describes experiments employing scanning ion conductance microscopy combined with FRET based cGMP sensors to show that NPR2 is uniformly located on the surface of cardiomyocytes whereas NPR1 is specifically located in T tubules. Based on this information, the authors suggest that the known effect of CNP/NPR2 but not ANP/BNP/NPR1 to inhibit ionotropy and stimulate lusitropic effects results from its more general location compared to the T tubular location of NPR1.

This is an interesting idea but seems more of a correlation than a fact. Furthermore, the whole idea of differential localization is solely based on indirect measurements of SCIM and FRET. Why not determine the location of NPR1 and NPR2 using antibody staining to directly demonstrate that these receptors are where you think they are? If there are not appropriate antibodies for these studies, is it possible to knock in receptors with antigenic tags to definitively determine the specific location of these receptors? The NPR2 KO mice could be used as a negative control for the immunochemical determination of NPR2.

More importantly, why is NPR1 but not NPR2 located to T tubules given that the primary amino acid sequence of these receptors is very similar. Can the authors determine what sequences are responsible for the unique localisations of the receptors?

Finally, one of the main things that ANP and BNP do through activation of NPR1 is to inhibit cardiac hypertrophy based on multiple KO and transgenic studies in mice. So how does NPR1 expression in

T tubules affect cardiomyocyte proliferation?

Reviewer #3 (Remarks to the Author):

The manuscript "Distinct submembrane localisation compartmentalises cardiac NPR1 and NPR2/cGMP signalling" by Subramanian et al. aims to ascertain the differential NPR1 and NPR2 signalling in cardiomyocytes. The results are clear. However, there are some points that need to be properly addressed.

General points:

1. The authors nicely demonstrated the differential NPR1 and NPR2 location within mouse cardiomyocytes. Thus, while NPR2 is uniformly localised through plasma membrane the NPR2 is concentrated T-tubules. The authors claim that this differential location is driven by lipid rafts. An important issue to be solve here is to determine whether PDE2 is restricted to T-tubules or, alternatively, could distribute elsewhere. Thus, the results presented so far do not demonstrated the restricted location of PDE2 into T-tubules. In addition, there are several important questions to be solved, for instance is PDE2 targeted to cholesterol rich domains? If so, which is the mechanism? Does M β CD treatment allows NPR1-mediated PLN phosphorylation? Does PDE2 physically interacts with NPR1 in T-tubules?

Specific points:

2. Which is the basal FRET signal in IBMX, BAY, cilostamide and tadalafil treated cardiomyocytes? This should be mentioned in the figure legends.

Reviewer #4 (Remarks to the Author):

This paper reports direct assessment of the distribution of two different natriuretic peptide receptors (NRP1 and NRP2) on the cell surface of cardiomyocytes. The authors capitalise on the unique methodology that they previously developed (Nikolaev et al, Science 2009). This is based on a complementary use of scanning ion conductance microscopy (SICM, to image membrane topology and deliver receptor agonists at nanoscale resolution) and Forster resonance energy transfer (FRET, to image receptor signalling). The major finding of the paper is that NRP1 receptor is localised to membrane invaginations (T-tubules) whilst NRP2 appears to be uniformly distributed throughout the cell surface. The authors argue that this allows spatial segregation of the intracellular cGMP signals produced by ANP/NRP1 (local) and CNP/NRP2 (global).

Although the paper potentially brings novel and important insights into the understanding of the cGMP signalling pathways in the heart, there are several shortcomings regarding quantitative details of the methods and controls that need to be addressed.

My main concern is how the authors estimate the local concentrations of ANP and CNP at the cell surface. I.e. what is the spatial resolution of their local application approach? What are the concentration gradients of the applied peptides near the application spot and how does this relate to the dissociation constants (K_ds) and the saturation levels of NRP1 and NRP2? For example is it possible that in the case of CNP application, the NRP2 receptors were saturated and therefore the authors could not pick up potential differences in the distribution of this receptor on the cardiomyocyte surface? Could the far-reaching CNP-induced FRET signals in Fig. 1d be a consequence of less local NRP2 activation caused by saturation? The authors do state that the actual ligand concentrations at individual receptors were well below 1 μ M (Extended Data Fig. 1), but it is not clear how this was estimated and how this concentration relates to NRP1/ANP and NRP2/CNP K_ds. A recent work by Babakinejad et. al (Anal. Chem. 2013) provides a quantitative

approach that allows one to estimate the concentration gradients in the vicinity of the SICM pipette. The authors need to consider using this methodology to substantiate their findings. In particular, they need to estimate the cell surface concentration gradients of the applied peptides and compare this to the size of T-tubules. They should also consider performing control experiments with reduced concentrations of CNP and ANP in the pipettes to test whether there are any issues with possible receptor saturation.

Other comments.

Fig. 1c,g.

It not clear why and how the authors used mixed ANOVA followed by Wald chi²-test as they only provide separate p-values for CNP and ANP experiments (for comparison of FRET signals between the T-tubule/Crest).

Fig. 1d,h. This is one of the major results of the paper, yet the authors only show two representative experiments. Later in the paper (Fig. 3b) the authors do compare the spatial differences between the signalling profiles for NRP1/APN/cGMP and NRP2/CPN/cGMP pathways, but it is not clear what was the number of individual experiments and the statistical significance. This need to be addressed.

Reviewer #1 (Remarks to the Author):

Subramanian and co-authors have submitted a manuscript in which they have investigated the subcellular localization of cGMP signaling downstream of natriuretic peptide receptors A and B (NPR-A, NPR-B). This was accomplished using ANP and CNP as activators of NPR-A and NPR-B, respectively, along with SICM and FRET based experimental approaches in isolated mouse ventricular myocytes.

The area of investigation is very interesting and important and has potential implications for how natriuretic peptides (NPs) can/should be used therapeutically for heart disease patients. Overall, the manuscript is well written and straightforward. The findings of the study potentially provide some explanation for differential effects of ANP and CNP on cardiac contractility.

While the study is generally well done, I have several comments that the authors should address in order to improve their study and strengthen their conclusions.

1. The use of SICM and FRET combines for a powerful experimental approach. Nevertheless, some clarification would be helpful. For example, the conclusions of the study depend on the ability to precisely deliver compounds to a single t-tubule or to the cell crests. Have the authors tried disrupting the t-tubules to confirm the selectivity of their approach (i.e. can you abolish the effects of ANP on cGMP production by disrupting the T-tubules chemically)?

We would like to thank the Reviewer for the overall positive judgement of our manuscript and for this helpful comment. As suggested we have performed additional FRET experiments to apply ANP onto cells with disrupted T-tubules. We have used dimethylformamide to osmotically disrupt T-tubules which is a standard *in vitro* approach for “chemical detubulation”. In detubulated cells, the response to ANP was indeed almost completely blunted (please see the new **Supplementary Figure 4a-c**). Since formamide detubulation is known to disrupt the connections of the intracellular T-tubular network to the outer sarcolemma with some remaining T-tubular openings on the surface, we also performed SICM/FRET experiments to apply ANP to T-tubular openings and crests of detubulated cells. Also in this case, we could see blunted ANP responses. We included these new data in the **new Supplementary Figure 4d-f**. Please, see also **new text on Page 5 (lines 5-10)**: “To confirm the selectivity of our approach for T-tubules, we performed acute detubulation experiments using dimethylformamide which causes osmotic loss of cell surface connections to the T-tubular system. Detubulation led to blunted ANP responses measured by FRET imaging under global ligand application (Supplementary Fig. 4a-c) or by SICM/FRET (Supplementary Fig. 4d-e).”

2. The authors also need to more comprehensively assess the dose dependence of NPs in their experiments. The doses used are unclear. The authors state that the pipette contained 100 μ M of ANP or CNP (a very high concentration) and suggest that actual concentrations at the receptors are ‘well below 1 μ M’ (which is still a concentration that is much higher than that in the circulation). I realize that local concentrations are likely higher than circulating levels, but it would be informative to present data in which the predicted concentrations at the receptors are in the low nanomolar range.

We thank the Reviewer for this very important comment. To estimate exact peptide concentrations at the membrane when applying from pipette filled with 100 μ M ANP or CNP, we have performed

two different experiments. First, we have used a fluorescent tracer (fluorescein) under the same conditions to measure the “dilution” of the ligand during application. As shown in the **new Supplementary Figure 2**, we indeed had at least a 1:100 dilution of fluorescein which means that the effective concentration at the membrane under our stimulation condition should be below 1 μ M. Second, following also the Reviewer 4 major comments, we used an established approach for computer-based finite simulations of ligand concentration gradients at the membrane. In this case we obtained values of 700 nM NP at the membrane during stimulation from pipette filled with 100 μ M NP – please see **new Supplementary Figure 3**. Moreover, we next measured concentration-response dependencies for our ANP and CNP effects by lowering ligand concentrations down to 70 nM with the same result re NPR1 and NPR2 localization (please see **new Figure 3a-b**). We have basically confirmed that decreasing NP concentrations down to more physiological values still shows the same receptor/cGMP localization pattern.

The respective new text passages can be found on page 4: “The peptide concentration at the surface was estimated by loading the pipette with 100 μ M of the fluorescent dye fluorescein and monitoring the fluorescence at the cell surface before and after delivery with different concentrations of fluorescein in the bath solution. These measurements showed that the concentration at the surface is approximately 1/100th of the pipette concentration (Supplementary Fig. 2). Finite element simulations using the program COMSOL Multiphysics was next performed to estimate the concentration profiles on the surface. The simulations showed that when applying from a pipette filled with 100 μ M ANP or CNP under bath perfusion, a maximum concentration of ~700 nM can be observed at the membrane with a relatively steep gradient which allows for a ~5-10-time concentration drop a distance of 2 μ m from the activated spot (Supplementary Fig. 3).” And in the middle of page 5 (second para): “To study concentration-response dependencies of SICM/FRET responses to CNP and ANP, we have lowered the applied NP concentrations by ~3.3 and 10 fold. In this case, CNP still induced comparable responses when applied to T-tubules and crests, while ANP responses remained confined to T-tubules (Fig. 3a,b). When compared to concentration-response dependences measured under global NP stimulation measured by FRET (Fig. 3c,d), around 10 times higher NP concentrations were needed to induce half-maximal response when VMs were stimulated locally in SICM/FRET experiments. This is probably due to local nature of ligand application in these experiments when only a small fraction of the total cellular NP receptor pool gets activated at a time”.

Redacted

3. If the authors conclusion is that ANP mediated production of cGMP is restricted by PDE2, why does ANP application in the presence of IBMX (which will inhibit PDE2 and other PDEs) produce a similar effect on cGMP signaling as seen in the absence of IBMX (supplemental Fig. 2G vs. Fig 1G). One would expect a larger effect of ANP following PDE inhibition.

We agree with the Reviewer that there was an obvious discrepancy in the first version of the manuscript, which was probably because of the limited n numbers. We have now performed two additional days of experiments including many more cells and carefully analyzing the data. Increasing the n numbers has now revealed clearly much stronger responses to ANP in presence of IBMX than to ANP alone. Please, see the **new version of the Figure 2b**

4. The authors state that NPR-A and NPR-B produce similar amounts of cGMP, but that the subcellular distribution of the two receptors is different. In fact, the authors data suggest that the amount of cGMP produced in the t-tubule compartment by CNP is much greater than ANP (Figure 1C and 1G). Please comment and, if necessary, revise the manuscript.

We thank the Reviewer for raising this point. Indeed, our FRET imaging data show stronger CNP effects as compared to ANP (Fig. 1, Fig. 3, Supplem. Fig 6, Götz KR et al. Circ Res 2014). These were all FRET experiments performed either with cytosolic or membrane targeted sensors which measure free cGMP concentration in the cytosol or at the plasma membrane. In the cytosol, CNP effects are usually bigger than ANP responses in mouse cells even after bath stimulation which might have something to do with compartmentation of NPR1/cGMP at the membrane or with lower NPR1 expression as compared to NPR2. Although, over the years, we have improved our cell isolation quality which led to stronger global cytosolic ANP signals, from very small ones published in Götz KR et al. Circ Res 2014 to descent ones such in the new supplementary fig. 6b-c. On the other hand, published radioimmunoassay data in human (e.g. Ref. 14 Dickey et al) and mouse heart lysates (Dickey et al 2007- Ref 17; Pierkes, Kuhn et al CVR 2002 and Holtwick, Kuhn et al, JCI 2003) show roughly similar total cGMP amount after NPR1 and NPR2 activation. We have added a sentence into the SICM/FRET results at the end of page 4: “ANP responses originating from T-tubules were generally lower than CNP signals which might be related to differences in cGMP compartmentation or receptor expression levels.”

5. The authors have used Npr2 knockout mice to confirm that the effects of ANP are mediated by NPR-A. please clarify the phenotype of these Npr2 knockout mice as global Npr2 knockouts are dwarfs, which may impact your study.

We thank the Reviewer for raising this important point. Indeed, we have used global Nrp2 knockouts for our study (because mice with cardiomyocyte specific deletion of Npr2 were not available). However, despite dwarfism the size and the structure of cardiomyocytes were indistinguishable from that of wildtype littermates. Neither could we detect any gross structural changes by SICM (e.g. see a representative scan in Fig. 2c) nor did we find differences in the size of myocytes, which we have now measured and included in the **new Supplementary Figure 5**. Therefore, we believe that the phenotype of these mice has no impact on our experiments and conclusions. We commented on that in the **new text on page 5, end of the first para**: “Since these global NPR2 knockout mice are dwarfs, we have measured the size of single VMs and found it not to be altered as compared to wildtype mice (Supplementary Fig. 5).”

6. The findings with M β CD and ANP are interesting. What are the effects of CNP in the presence of this compound? It would also be helpful to present representative data in addition to the summary bar graphs (Supplemental Figure 3).

We thank the Reviewer for this helpful suggestion. We have now included representative traces for ANP in M β CD treated cells. Please, see the **new Supplemental Figure 3b**. We have also conducted new SICM/FRET experiments using CNP in M β CD treated cells – these new data are included in the **new Supplemental Figure 3 d-f**. CNP responses were not largely affected by this treatment. Please, see also new text on Page 6, second para: “In contrast, the amplitude and localisation of the NPR2/cGMP were not affected by methyl- β -cyclodextrin treatment (Supplementary Fig. 6d-e).”

7. Although inhibition of PDE2 does appear to facilitate farther reaching ANP mediated cGMP signals, they are still substantially smaller than those observed with CNP. Is it possible that other PDEs (or molecules other than PDEs) are also involved? Or is this reflective of your data that seem to suggest CNP produces more cGMP than ANP? Please comment.

We believe that this is mostly reflective of our data pointing to different overall magnitudes of ANP and CNP responses (see answer to point 4 above). However, we do not exclude that other mechanisms might be involved.

8. The authors need to be specific throughout their manuscript that they are using ventricular myocytes. Using ‘cardiomyocytes’ is not clear enough. This is important because it is conceivable that very different findings would be observed in atrial or sinoatrial node myocytes, which have different cellular morphology and different patterns in the t-tubular network. Indeed, studies have shown, for example, that BNP and CNP have very similar effects on heart rate and ion currents in the sinoatrial node.

Thank you for raising this point. We have now changed the term “cardiomyocytes (CMs)” to “ventricular myocytes (VMs)” throughout the manuscript.

9. The authors may be missing an important contribution of PDE3 in the interpretation of their findings. cGMP produced downstream of NPRs can inhibit PDE3 (including at lower concentrations than those required to activate PDE2), which can result in an elevation in cAMP-PKA signaling. The authors show some effects of PDE3 inhibition in their results and this could also impact on, for example, phosphorylation of PLB and other targets. The authors should address this in their discussion.

Indeed, especially after beta-adrenergic stimulation, CNP- or ANP-induced cGMP can participate in a positive cGMP/cAMP “cross-talk”, increasing cAMP levels via PDE3 inhibition. We could also visualize it by FRET – see e.g. Fig. 7 in Götz KR et al Circ Res. 2014 (Ref. 18). However, Frantz et al. Eur Heart J 2013 have shown that CNP-induced PLB phosphorylation (in mouse VMs without beta-adrenergic prestimulation, under similar conditions as in our current manuscript) is completely abolished in cGKI knockout cells. Therefore, we think the in our case, this cross-talk mechanism via cAMP-PKA signaling

is of a lower impact. We have added these considerations into discussion on page 8: “It is well documented that CNP stimulation of VMs leads to PLN phosphorylation which is completely abolished in cGK knockout cells²⁵. In addition, NP-stimulated cGMP can participate in the so-called positive cGMP-to-cAMP cross-talk, inhibiting PDE3 and increasing cAMP levels which can further enhance electromechanical coupling, especially under concomitant beta-adrenergic stimulation^{19,28}.”

Minor comments

1. The authors have oversimplified the introduction (page 3) by stating that ANP is an atrial peptide and BNP is a ventricular peptide. In fact, ANP and BNP are both present in atrial granules in the normal heart. Similarly, particularly in heart failure, the ventricles produce BNP and ANP. The authors should clarify their introduction and provide appropriate citations.

We thank the Reviewer for this minor point and apologize for having oversimplified NP physiology. We have rephrased this sentence to “ANP and BNP, which are both produced by stretched atria or by diseased VMs, activate NPR1 and can counteract pathological cardiac hypertrophy.” and provided appropriate review citations. Please, see updated text on page 3.

2. Please present a comprehensive panel of echocardiography findings (Supplemental Figure 4). The authors should include chamber dimensions and wall thicknesses in systole and diastole along with heart rate etc.

We thank the Reviewer for this helpful suggestion. We have now included all above mentioned echo parameters into the Supplemental Figure 9.

3. The authors should consider moving some/all of the supplemental data into the main paper as there is plenty of room to do so and the results in these figures are important.

Many thanks. We have now moved two figures into the main paper – see new Fig 2 and Fig 3 which now seem to provide a good logical flow. We can eventually move some more supplements to the main figures if needed at Editor’s discretion.

Reviewer #2 (Remarks to the Author):

The manuscript titled "Distinct submembrane localisation compartmentalises cardiac NPR1 and NPR2/cGMP signalling by Subramanian et al. describes experiments employing scanning ion conductance microscopy combined with FRET based cGMP sensors to show that NPR2 is uniformly located on the surface of cardiomyocytes whereas NPR1 is specifically located in T tubules. Based on this information, the authors suggest that the known effect of CNP/NPR2 but not ANP/BNP/NPR1 to inhibit ionotropy and stimulate lusitropic effects results from its more general location compared to the T tubular location of NPR1.

This is an interesting idea but seems more of a correlation than a fact. Furthermore, the whole idea of differential localization is solely based on indirect measurements of SICM and FRET. Why not determine the location of NPR1 and NPR2 using antibody staining to directly demonstrate that these receptors are where you think they are? If there are not appropriate antibodies for these studies, is it possible to knock in receptors with antigenic tags to definitively determine the specific location of these receptors? The NPR2 KO mice could be used as a negative control for the immunochemical determination of NPR2.

We thank the Reviewer for this important and valuable suggestion. What we intend with our study using SICM/FRET is to provide a readout for functional NP receptor localization in myocytes which is important for the physiology. Of course, one can imagine other pools of receptors located at different membrane structures which could be detected by other approaches. However, we firmly believe that the particular strength of our approach (as compared to above mentioned techniques) is that we can not only show where the receptors are localized but also if and how it functions. In this regard a "correlation" with function i.e. the ability to produce cGMP is in our opinion, of an additional positive rather than negative value. Indeed, an alternative method for physical localization of receptors would be nice. As you mentioned, the sensitivity of NPR1 and NPR2 antibodies is indeed too poor. The best available NPR1 antibody cannot detect endogenous receptor in isolated myocytes even in Western blot (only in NPR1 transgenic hearts – see Ref. 38 Klaiber et al.), although we know it is there based on mRNA expression and on numerous functional studies including those in cardiomyocyte-specific KO mice. The best available NPR2 antibody works for immunoblot (see Ref. 20 Ter-Avetisyan et al.) and not for immunostaining cardiomyocytes, so that classical IF or EM studies with endogenous receptors are not possible with available tools.

Generating an affinity tagged knockin mouse models for NPR1 and NPR2 could be indeed an interesting possibility. Although using CRISPR/Cas9 it is possible to generate these models relatively/seemingly fast, their proper development and characterization under perfect scenario will not take less than a year, so we believe it will exceed the time allocated for the revision and might be out of scope for this particular study. In addition, there are several other difficulties using this approach for NP receptors (as compared e.g. to G-protein coupled receptors). For example, the exact position of the antigenic tag is not trivial, since it should be not just on N-terminus but somewhere between the N-terminal signal peptide and the extracellular domain, there are several possibilities there which should be properly tested before making mice. Moreover, my colleagues in Minnesota and Connecticut who are very experienced and renown experts in NPR field have been working on exactly this project (knockin mouse generation) since already 2 years. Their first attempt with Flag-tagged NPR2 was unsuccessful, while the HA-tagged constructs are under development and characterization. We do not want to merely replicate their work, so we decided to concentrate on

SICM/FRET approach for now and let other further studies confirm our finding at the level of physical receptor localization.

More importantly, why is NPR1 but not NPR2 located to T tubules given that the primary amino acid sequence of these receptors is very similar. Can the authors determine what sequences are responsible for the unique localisations of the receptors?

We would like to thank the Reviewer for raising this very helpful point. As suggested, we looked more closely which sequences of the receptors might be responsible to differential localization. Our cholesterol depletion data (Supplem. Fig. 6) indicate that there might be differences in interaction with caveolin which is also enriched in T-tubuli. When looking at NPR1 and NPR2 protein sequences, we could observe that despite overall high sequence similarity, transmembrane domains of both receptors are very different (**new Supplementary Fig. 7**). Therefore, we to this end, tested the hypothesis that transmembrane domains of NPR1 and NPR2 can differentially interact with caveolin and lipid rafts using a well-established method called FRAP microscopy which measures molecular mobility of the receptor. We expressed YFP-tagged NPR1 and NPR2 in HEK293 cells and analyzed their mobility with and without M β CD treatment by FRAP. We found that NPR1 but not NPR2 was sensitive to M β CD treatment (showing a decrease in mobility which is well documented for proteins associated with caveolin-rich membrane domains or lipid rafts), suggesting its association with caveolin-rich membrane microdomains and/or lipid rafts. When we exchanged the transmembrane domain of NPR1 with that of NPR2, the sensitivity to M β CD treatment was lost. Conversely, it was gained when exchanging this domains in NPR2 by the transmembrane domain from NPR1 (**new Supplementary Fig. 8**).

We described these data in the new text passage on page 6 and 7: “To test which structural features of NPR1 might be responsible for its interaction with caveolin-rich membrane domains, we developed receptor constructs in which NPR1 and NPR2 were fused to enhanced yellow fluorescent protein (EYFP) to study receptor mobility by fluorescence recovery after photobleaching (FRAP) in transfected HEK293A cells. Despite overall high similarity, NPR1 and NPR2 protein sequences differ considerably in the transmembrane domain (TMD), so that we decided to clone chimeric receptors with swapped TMDs (Supplementary Fig. 7). FRAP microscopy revealed that NPR1-EYFP but not NPR2-EYFP mobility was, as expected, sensitive to M β CD treatment suggesting its localisation in lipid rafts or caveolin-rich domains (Supplementary Fig. 8a,c,d,f). A decrease in mobility after M β CD is compatible with the behavior of proteins located in these membrane compartments. Interestingly, exchange of NPR1 TMD with that of NPR2 completely abolished M β CD sensitivity (Supplementary Fig. 8b,c), suggesting that TMD is important for localisation of NPR1 to caveolin-rich domains. Conversely, exchange of NPR2 TMD with the of NPR1 has led to a M β CD induced decrease in mobility (Supplementary Fig. 8e,f). Different direction of M β CD effect indicates that isolated TMD might be just one of several structural features required for proper receptor localisation and mobility.”

Finally, one of the main things that ANP and BNP do through activation of NPR1 is to inhibit cardiac hypertrophy based on multiple KO and transgenic studies in mice. So how does NPR1 expression in T tubules affect cardiomyocyte proliferation?

Thank you for raising this important point! One of the mechanisms how NPR1 can regulate cardiomyocyte hypertrophy/proliferation is via TRPC channels. It is well established that TRPC channels are localized in T-tubules. Some years ago, we have also provided evidence that NPR1 and

TRPC3/6 form a stable complex in myocytes and affect each other's activity (ref. 38), showing that this mechanism is also involved in hypertrophy regulation. It is tempting to speculate that multimolecular NPR1/TRPC complex in the T-tubules might be an important module. We have included these considerations in our discussion on page 9: "Interestingly, ANP/NPR1 pathway has been shown to functionally interact with transient receptor potential subfamily C (TRPC) channels to prevent cardiac hypertrophy²⁹ via cGKI mediated phosphorylation which inhibits channel activity³⁰. It is well documented that TRPC channels are located in T-tubules of VMs^{31,32} and can even form a stable molecular complex with NPR1 (Ref 33). So it is tempting to speculate that localisation of NPR1 in T-tubules promotes its molecular and functional association with TRPC channels to mediate antihypertrophic effects of ANP and BNP."

Reviewer #3 (Remarks to the Author):

The manuscript “Distinct submembrane localisation compartmentalises cardiac NPR1 and NPR2/cGMP signalling” by Subramanian et al. aims to ascertain the differential NPR1 and NPR2 signalling in cardiomyocytes. The results are clear. However, there are some points that need to be properly addressed.

General points:

1. The authors nicely demonstrated the differential NPR1 and NPR2 location within mouse cardiomyocytes. Thus, while NPR2 is uniformly localised through plasma membrane the NPR2 is concentrated T-tubules. The authors claim that this differential location is driven by lipid rafts. An important issue to be solve here is to determine whether PDE2 is restricted to T-tubules or, alternatively, could distribute elsewhere. Thus, the results presented so far do not demonstrated the restricted location of PDE2 into T-tubules. In addition, there are several important questions to be solved, for instance is PDE2 targeted to cholesterol rich domains? If so, which is the mechanism? Does MβCD treatment allows NPR1-mediated PLN phosphorylation? Does PDE2 physically interacts with NPR1 in T-tubules?

We thank the Reviewer for the overall positive assessment of our manuscript and for helpful critiques and suggestions. We have performed new experiments to address these questions point-by-point:

Does MβCD treatment allows NPR1-mediated PLN phosphorylation?

Here we performed immunoblots to look at PLN phosphorylation in the presence of MβCD. As shown by some older studies, MβCD treatment at 1 mM alone could already increase basal phosphorylation. However, we found that 0.5 mM MβCD does not have such a dramatic effect on basal phosphorylation but can unmask ANP-induced PLN phosphorylation which suggests that the confinement of NRP1 in lipid rafts indeed should play a major role in the compartmentation of NPR1/cGMP signals. We have included these data in the **new Supplementary Fig. 11**

Does PDE2 physically interacts with NPR1 in T-tubules?

We first checked whether NRP1 and PDE2A can form a stable molecular complex using Co-IP experiments with Flag-NRP1 and PDE2A3 in both directions in HEK293 cells. As shown in the **new Supplementary Fig. 10**, this was not the case. Based on these data and on our previous work (Perera RK et al. Circ Res 2015), we believe that PDE2 is located rather in close proximity to NPR1 in a similar membrane microdomain than directly interacts with this receptor.

Is PDE2 targeted to cholesterol rich domains? If so, which is the mechanism?

Although the mechanisms of subcellular targeting of PDE2A are not completely understood, we know from the literature that in the brain, PDE2A3 is indeed targeted to cellular membranes by dual acylation, i.e. myristoylation at its N-terminus (Russwurm C et al. JBC 2009). We have addressed our data in text at the end of page 8: “Although PDE2 does not seem to directly interact with NPR1 based on co-immunoprecipitation experiments (Supplementary Figure 10), ANP-induced PLN phosphorylation could be unmasked by incubating VMs with MβCD (Supplementary Figure 11),

suggesting that proper localisation of NPR1 to caveolin-rich membrane domains including T-tubuli is important for compartmentation of its cGMP signals.”, and this PDE2 localization mechanism in the **discussion on page 9**: “The exact mechanisms responsible for subcellular PDE2 localisation in VMs are not well understood, although one study in performed in neurons, showed that dual myristoylation at the N-terminus of PDE2A3 is responsible for its membrane targeting²⁷.”

Specific points:

2. Which is the basal FRET signal in IBMX, BAY, cilostamide and tadalafil treated cardiomyocytes? This should be mentioned in the figure legends.

Thank you for this comment. We have amended the respective Figure legends (see Fig. 4) as follows: “In c,d and f, Sapp/RFP FRET ratio before addition of PDE inhibitors (at basal state of after NP stimulation) was normalised on 1 to calculate % changes in FRET ratio and to facilitate the comparison between various conditions.”

Reviewer #4 (Remarks to the Author):

This paper reports direct assessment of the distribution of two different natriuretic peptide receptors (NRP1 and NRP2) on the cell surface of cardiomyocytes. The authors capitalize on the unique methodology that they previously developed (Nikolaev et al, Science 2009). This is based on a complementary use of scanning ion conductance microscopy (SICM, to image membrane topology and deliver receptor agonists at nanoscale resolution) and Forster resonance energy transfer (FRET, to image receptor signalling). The major finding of the paper is that NRP1 receptor is localised to membrane invaginations (T-tubules) whilst NRP2 appears to be uniformly distributed throughout the cell surface. The authors argue that this allows spatial segregation of the intracellular cGMP signals produced by ANP/NRP1 (local) and CNP/NRP2 (global).

Although the paper potentially brings novel and important insights into the understanding of the cGMP signalling pathways in the heart, there are several shortcomings regarding quantitative details of the methods and controls that need to be addressed.

My main concern is how the authors estimate the local concentrations of CNP and ANP at the cell surface. I.e. what is the spatial resolution of their local application approach? What are the concentration gradients of the applied peptides near the application spot and how does this relate to the dissociation constants (Kds) and the saturation levels of NRP1 and NRP2? For example is it possible that in the case of CNP application, the NRP2 receptors were saturated and therefore the authors could not pick up potential differences in the distribution of this receptor on the cardiomyocyte surface? Could the far-reaching CNP-induced FRET signals in Fig. 1d be a consequence of less local NRP2 activation caused by saturation? The authors do state that the actual ligand concentrations at individual receptors were well below 1 μ M (Extended Data Fig. 1), but it is not clear how this was estimated and how this concentration relates to NRP1/APN and NRP2/CNP Kds. A recent work by Babakinejad et. al (Anal. Chem. 2013) provides a quantitative approach that allows one to estimate the concentration gradients in the vicinity of the SICM pipette. The authors need to consider using this methodology to substantiate their findings. In particular, they need to estimate the cell surface concentration gradients of the applied peptides and compare this to the size of T-tubules. They should also consider performing control experiments with reduced concentrations of CNP and ANP in the pipettes to test whether there are any issues with possible receptor saturation.

We would like to thank the Reviewer for the overall positive comment on our manuscript and very important suggestions on how to improve the quantitative details of the study. Following these helpful comments, we performed computer based simulation following the suggested approach described in Babakinejad et. al (Anal. Chem. 2013).

Firstly, to measure the overall dilution of the ligand during application, we have used a fluorescent tracer (fluorescein) under the same application conditions as used in our SICM/FRET experiments. As shown in the **new Supplementary Figure 2**, we indeed had an at least a 1:100 dilution of fluorescein which showed no more than 1 μ M at the membrane under our stimulation condition. Secondly, we turned to the Babakinejad et. al approach to compute ligand concentration gradients at the membrane. In this case, we obtained peak values of 700 nM NP at the membrane during stimulation from pipette filled with 100 μ M NP – please see **new Supplementary Figure 3**. When using superfusion on the bath, as in all our experiments, the size of the activated membrane spot could be significantly reduced, so that a relatively steep gradient which allows for a \sim 10-times

concentration drop at the distance of 1 μm from the activated spot could be achieved. This should be sufficient to selectively activate a crest area without much affecting the neighboring T-tubules and conversely, to activate a single T-tubules without much affecting other T-tubules and the crest.

Finally, we next measured concentration-response dependencies for our ANP and CNP effects by lowering ligand concentrations down to 70 nM with the same result re NPR1 and NPR2 localization (please see **new Figure 3a-b**). EC50 values measured by FRET upon global bath stimulation were around 10 nM (please see **new Figure 3c-d**) The Kd of receptors are in the range of 1 nM at room temperature (Koeller KJ et al, Mol Cell Biol 1992). Please, see also our considerations about neprilysin - response to Reviewer 1, point 2, last para.

The respective new text passages can be found on page 4: “The peptide concentration at the surface was estimated by loading the pipette with 100 μM of the fluorescent dye fluorescein and monitoring the fluorescence at the cell surface before and after delivery with different concentrations of fluorescein in the bath solution. These measurements showed that the concentration at the surface is approximately 1/100th of the pipette concentration (Supplementary Fig. 2). Finite element simulations using the program COMSOL Multiphysics was next performed to estimate the concentration profiles on the surface. The simulations showed that when applying from a pipette filled with 100 μM ANP or CNP under bath perfusion, a maximum concentration of ~ 700 nM can be observed at the membrane with a relatively steep gradient which allows for a ~ 5 -10-time concentration drop a distance of 2 μm from the activated spot (Supplementary Fig. 3).” And in the middle of page 5 (second para): “To study concentration-response dependencies of SICM/FRET responses to CNP and ANP, we have lowered the applied NP concentrations by ~ 3.3 and 10 fold. In this case, CNP still induced comparable responses when applied to T-tubules and crests, while ANP responses remained confined to T-tubules (Fig. 3a,b). When compared to concentration-response dependences measured under global NP stimulation measured by FRET (Fig. 3c,d), around 10 times higher NP concentrations were needed to induce half-maximal response when VMs were stimulated locally in SICM/FRET experiments. This is probably due to local nature of ligand application in these experiments when only a small fraction of the total cellular NP receptor pool gets activated at a time”.

Other comments.

Fig. 1c,g.

It not clear why and how the authors used mixed ANOVA followed by Wald chi2-test as they only provide separate p-values for CNP and ANP experiments (for comparison of FRET signals between the T-tuble/Crest).

We have used this type of statistical test in all cases where multiple cells from several animals were analyzed, so that a “nested” type of ANOVA was necessary to account for inter- and intra-animal variability. This analysis provides separate p values for individual comparison of groups of cells (originating from different animals, typically several cells from each animal were included) which were treated either on T-tubules or crests. These were not the same cells treated at both locations which would require a paired type of statistical test.

Fig. 1d,h. This is one of the major results of the paper, yet the authors only show two representative experiments. Later in the paper (Fig. 3b) the authors do compare the spatial

differences between the signalling profiles for NRP1/APN/cGMP and NRP2/CPN/cGMP pathways, but it is not clear what was the number of individual experiments and the statistical significance. This need to be addressed.

Indeed, in Figure 1d and h we only showed representative experiments, while the quantification of all such experiments is included in Figure 3b (now Figure 5b). We apologize that we have not clearly mentioned it in the respective figure legend and did not provide n numbers. We have now amended the legend to figure 5b to include this information: “b includes also the quantification of experiments shown in Figures 1d and h. Shown are means \pm s.e.m. 5 cells from 3 mice per condition were analysed. Differences between all three conditions were significant at $p=0.05$ for each individual distance value using mixed ANOVA followed by Wald χ^2 -test”

REVIEWERS' COMMENTS:

Reviewer #1 (Remarks to the Author):

The authors have submitted a revised manuscript in which they have responded to my comments on their first submission in a meaningful, thoughtful way, including with the addition of additional experimental results. This is certainly appreciated and the manuscript is much improved.

I do have one additional comment that arises based on the addition of new data in which NPR chimeras have been studied in conjunction with application of MbCD (for cholesterol depletion). The authors have argued that MbCD application causes redistribution of Npr1 from t-tubules to crests and show that MbCD has effects that occur in different directions based on the chimera being studied. Unfortunately, this is confusing and has, in some respects, decreased the clarity of the results because it is unclear why the effects occur in different directions. Cholesterol depletion would be anticipated to disrupt caveolae, which are mostly located on the plasma membrane rather than in t-tubules. Can the authors comment on this? Does the SICM/FRET approach distinguish between 'depressions' in the plasma membrane that are representative of openings of t-tubules vs. caveolae? This is important as NPR-A has been identified in caveolae in some tissues.

A minor comment is that it is more conventional to refer to the receptors as NPR-A and NPR-B rather than Npr1 and Npr2. They authors may consider adopting this in their manuscript.

Reviewer #2 (Remarks to the Author):

The revised manuscript by Subramanian et al. is substantially improved and tells a convincing story in its current form. Although, it would have benefitted from the suggested immunolocalization studies using mice expressing N-terminally tagged forms of NPR2, these experiments have proven more difficult than originally anticipated by all members of the NPR2 research community. Hence, it is not reasonable to delay the publication of this informative article until these experiments are completed. Furthermore, this article contains an extensive amount of new data that conveys information of immediate interest to the natriuretic peptide and guanylyl cyclase communities, and therefore, publication should be expedited, not delayed. This reviewer finds the newly added information regarding the unique properties of the transmembrane domains of NPR1 and NPR2 to be one of the more unique findings of the manuscript. Hence, if possible, the authors may consider moving supplemental figures 7 and 8 to the main results section of the manuscript. Finally, "for" is misspelled on line 12 of page 6.

Lincoln R. Potter

Reviewer #3 (Remarks to the Author):

The authors properly addressed all the questions raised by this referee, thus the manuscript deserves publication.

Reviewer #4 (Remarks to the Author):

In the second revision, the authors have conducted the additional experiments and analysis suggested by the reviewers. The paper has substantially improved and now the conclusions are fully substantiated by the experimental data. In my opinion, the use of the innovative original

methodology allowed the authors to bring novel important insights into the understanding of the cGMP signalling pathways in the heart. I have no further reservations and can now recommend this manuscript for publication.

RESPONSES TO REVIEWERS' COMMENTS

We would like to thank the Reviewers for their extremely positive feedback to the revised version of our manuscript! We did our best to address the remaining issues as discussed in point-by-point responses below.

Reviewer #1 (Remarks to the Author):

I do have one additional comment that arises based on the addition of new data in which NPR chimeras have been studied in conjunction with application of MbCD (for cholesterol depletion). The authors have argued that MbCD application causes redistribution of Npr1 from t-tubules to crests and show that MbCD has effects that occur in different directions based on the chimera being studied. Unfortunately, this is confusing and has, in some respects, decreased the clarity of the results because it is unclear why the effects occur in different directions. Cholesterol depletion would be anticipated to disrupt caveolae, which are mostly located on the plasma membrane rather than in t-tubules. Can the authors comment on this? Does the SICM/FRET approach distinguish between 'depressions' in the plasma membrane that are representative of openings of t-tubules vs. caveolae? This is important as NPR-A has been identified in caveolae in some tissues.

Indeed, we fully agree with the Reviewer that different directions of MbCD effect in wildtype NPR1 and in the NPR2/1 mutant are somewhat confusing. We have no other good explanations to this discrepancy, apart from two possibilities. One is that TMD one of the important but not the only one domain which is responsible for the targeting of NPR1 to the lipid rafts. It might act in combination with some other features of this receptor. This is what we tried to say when discussing this result in the text. On the other hand, we firmly believe that one cannot easily compare the membrane structures of HEK cells with that of adult cardiomyocytes, the latter ones having T-tubules and caveolar at the outer membrane. Since adult myocytes are not easily transfectable with plasmids, we used HEK cells to transfect chimeric constructs and study the basic molecular features which affect the mobility of the receptor (on the yes or no MbCD sensitivity basis). It is possible that due to unique membrane microdomain composition, the same constructs would behave differently in adult myocytes. We have added this point and rephrased our text on page 7 as follows:

“Conversely, exchange of NPR2 TMD with the of NPR1 has regained MbCD sensitivity but led to a decrease in mobility (Figure 4e,f). Different direction of MbCD effect might indicate that isolated TMD could be just one of several structural features required for proper receptor localisation and mobility. On the other hand, membrane microdomain structures of HEK293A cells differ from those of myocytes which might affect the behavior of such constructs”.

Our SICM/FRET approach at the lateral resolution which we use (40-50 nm) can distinguish between T-tubules and outer membrane / crests but cannot nail the crest down to individual caveolae. They are possible to image by the so-called high resolution SICM which uses 10-20

nm pipettes (see e.g. Figure 1D in Wright, Nikolaev et al, JMCC 2014, PMID 24345421). However, it is not possible to deliver ligand through such a small pipette because of the size.

A minor comment is that it is more conventional to refer to the receptors as NPR-A and NPR-B rather than Npr1 and Npr2. They authors may consider adopting this in their manuscript.

When initially writing the manuscript, we checked *Nature* guidelines for nomenclature which suggests using abbreviations corresponding to the gene names. For this reason we decided to use NPR1 and NPR2 instead of GC-A and GC-B which we normally use. We would be happy to do it either way, so I would probably leave it to the Editor to decide which nomenclature we should use.

Reviewer #2 (Remarks to the Author):

This reviewer finds the newly added information regarding the unique properties of the transmembrane domains of NPR1 and NPR2 to be one of the more unique finding of the manuscript. Hence, if possible, the authors may consider moving supplemental figures 7 and 8 to the main results section of the manuscript.

Thank you very much for this suggestion. We have now moved the FRAP data from the supplements into the new Figure 5.

Finally, "for" is misspelled on line 12 of page 6.

This typo has been fixed.